# Lung clearance index in healthy volunteers, measured using a novel portable system with a closed circuit wash-in

**Alex R. Horsley**[1,2]*, **Amnah Alrumuh**[3,4], **Brooke Bianco**[2,5], **Katie Bayfield**[5], **Joanne Tomlinson**[4], **Andrew Jones**[1,2], **Anirban Maitra**[6], **Steve Cunningham**[7], **Jaclyn Smith**[1], **Catherine Fullwood**[8,9], **Anand Pandyan**[3], **Francis J. Gilchrist**[3,4]

**1** Division of Infection, Immunity & Respiratory Medicine, Faculty of Biology Medicine and Health, University of Manchester, Manchester, United Kingdom, **2** Manchester Adult CF Centre, Manchester University NHS Foundation Trust, Manchester, United Kingdom, **3** Institute of Applied Clinical Science, Keele University, Newcastle-under-Lyme, United Kingdom, **4** Royal Stoke University Hospital, University Hospitals of North Midlands NHS Trust, Stoke-on-Trent, United Kingdom, **5** NIHR Manchester Clinical Research Facility, Manchester, United Kingdom, **6** Royal Manchester Children's Hospital, Manchester University NHS Foundation Trust, Manchester, United Kingdom, **7** Centre for Inflammation Research, University of Edinburgh, Edinburgh, United Kingdom, **8** Research and Innovation, Manchester University NHS Foundation Trust, Manchester, United Kingdom, **9** Centre for Biostatistics, Faculty of Biology Medicine and Health, University of Manchester, Manchester, United Kingdom

* alexander.horsley@manchester.ac.uk

**Data Availability Statement:** Data cannot be shared publicly because this was not explicitly requested in the volunteer consent. Data are

## Abstract

### Introduction

Lung clearance index (LCI) is a sensitive measure of early lung disease, but adoption into clinical practice has been slow. Challenges include the time taken to perform each test. We recently described a closed-circuit inert gas wash-in method that reduces overall testing time by decreasing the time to equilibration. The aim of this study was to define a normative range of LCI in healthy adults and children derived using this method. We were also interested in the feasibility of using this system to measure LCI in a community setting.

### Methods

LCI was assessed in healthy volunteers at three hospital sites and in two local primary schools. Volunteers completed three washout repeats at a single visit using the closed circuit wash-in method (0.2% $SF_6$ wash-in tracer gas to equilibrium, room air washout).

### Results

160 adult and paediatric subjects successfully completed LCI assessment (95%) (100 in hospital, 60 in primary schools). Median coefficient of variation was 3.4% for LCI repeats and 4.3% for FRC. Mean (SD) LCI for the analysis cohort (n = 53, age 5–39 years) was 6.10 (0.42), making the upper limit of normal LCI 6.8. There was no relationship between LCI and multiple demographic variables. Median (interquartile range) total test time was 18.7 (16.0–22.5) minutes.

available from the corresponding author or from the Clinical Research Facility at Wythenshawe Hospital (email angela.kelsall@mft.nhs.uk), and would be made available following application and approval from Lancaster NHS Research Ethics Committee.

**Funding:** This study was funded by a National Institute for Health Research (www.nihr.ac.uk) Clinician Scientist award to AH (NIHRCS12-013). The funders had no role in study design, data collection and analysis, decision to publish, or preparation of the manuscript.

**Competing interests:** AH reports a collaboration agreement with Innovision ApS. All other authors report no conflicts of interest. This does not alter our adherence to PLOS ONE policies on sharing data and materials.

## Conclusion

The closed circuit method of LCI measurement can be successfully and reproducibly measured in healthy volunteers, including in out-of-hospital settings. Normal range appears stable up to 39 years. With few subjects older than 40 years, further work is required to define the normal limits above this age.

## Introduction

Multiple breath washout (MBW) is now a well-researched technique to assess lung physiology. Supported by international guidelines, it has been used as an endpoint in therapeutic trials and is now being measured for clinical use in a handful of specialist units[1–5]. Most of the clinical and research use of MBW has been in the field of cystic fibrosis (CF) with lung clearance index (LCI) being the best described derived outcome[6]. There are particular perceived advantages of using LCI over the more traditional $FEV_1$ in the CF population: it appears to be highly sensitive to early disease[7], is reproducible[8], and is sensitive to clinically meaningful changes[9, 10].

An important characteristic noted in early research of LCI was that it appeared to have a stable upper limit of normal in younger patients[11, 12]. A stable range of normal means that any disease-related change over time can easily be identified. The evolution of the technique however has revealed that different systems produce different measures of LCI and functional residual capacity (FRC) due to differences in tracer gas (e.g. nitrogen or sulphur hexafluoride ($SF_6$)) [13, 14] as well as more subtle changes in equipment deadspace[15], gas analyser response time and signal alignment [16, 17]. Studies in older adults using nitrogen washout have also indicated that, along with other markers of lung function and airway elasticity, there is an age-related increase in markers of ventilation heterogeneity, including nitrogen-LCI[18, 19].

One of the major factors limiting the wider adoption of LCI is the time taken to complete a test. The final result is derived from an average of three repeated tests, and additional tests may be required in younger subjects or in case of poor quality tests in order to produce repeatable results. Longer physiology testing times disrupt clinic scheduling and may not be tolerated by younger children. When researchers tried to limit the test time to 20 minutes using nitrogen washout in children with CF too few children obtained successful results, with the required minimum of two or more repeats achieved in only 40% of participants[20]. To enable the wider clinical adoption of LCI would require reduction in time taken to perform the test and increased flexibility in where testing occurs. Closed circuit wash-in methodologies could achieve these outcomes and for that reason our centre has developed and refined this methodology. This involves the wash-in of tracer gas from a sealed bag of air enriched with $O_2$ and $SF_6$ [21], enabling a higher concentration of tracer gas ($SF_6$) to be drawn into the lungs at the start of wash-in. The final mixed alveolar $SF_6$ concentration is therefore reached more quickly than with a conventional open circuit wash-in [21, 22]. Since the $O_2/SF_6$ mix is supplied from a small cylinder within the device, the system is portable and can be mounted on a medical cart with its own battery power supply. This enables it to be taken to patients in hospital, and opens the possibility of performing measurements in community settings.

The Innocor™ analyser has been available to measure LCI for over a decade, but the closed circuit system we have developed and report here differs in terms of deadspace and response time from that originally described[11]. The washout analysis has also evolved, and now

incorporates consideration of re-inspired tracer. In preparation for its use in a longitudinal clinical study, the aims of this study were to:

1. Evaluate the range of LCI values seen in healthy subjects.

2. Calculate the upper limits of normal for LCI using this device, and define the relationship between LCI and key demographic variables (subject height and weight/bmi, age and gender).

3. Assess the feasibility of using the closed circuit device to perform measurements in a community setting.

## Methods

### Study design and recruitment

This was an observational study of healthy subjects completing LCI assessment on a single occasion at three hospitals and two local schools (one primary school and one secondary school). The hospitals were: Wythenshawe Hospital, Manchester, UK (WH); Royal Manchester Children's Hospital (RMCH); and Royal Stoke University Hospital (RSUH). Adult subjects were only assessed at WH, whereas paediatric subjects were assessed at all three sites. Measurements were performed on one of three identically set up and calibrated Innocor gas analysers, using a closed circuit wash-in. These were mounted on battery-supplied medical carts to allow portable measurements within hospital (see Fig 1). The device used at RSUH was also transported in a protective case and taken to two local schools in Staffordshire to measure LCI in the community. There were three different teams responsible for LCI assessment (one covering RMCH and WH, one for RSUH, and one for community measurements), but all were trained and supervised by the same researcher to the same training standard.

Subjects were recruited by advertisement in hospitals from amongst staff and patient relatives, from an outpatient fracture clinic (RSUH) and by contact through the schools. Subjects were over the age of 5 years, were non-smokers or ex-smokers of >6months with less than 5 pack year smoking history, with no history of asthma or wheeze requiring any inhaler use in the last 12m. Additional exclusions included history of cardiac disease, pertussis, tuberculosis, or prematurity (<34 weeks). All participants provided assent, and parents and adult volunteers provided signed informed consent. This study was approved by the Lancaster NHS Research Ethics Committee (study reference 14/NW/1195) and the Keele University Ethics Committee. Participants were recruited between December 2014 and November 2018.

### Study assessment

Multiple breath washout was performed using a closed circuit Innocor™ system (PulmoTrace ApS, Glamsberg, Denmark), as previously described [21]. Participants wore a nose-clip and breathed through the apparatus using a mouthpiece. Wash-in was performed from a sealed bag filled with a mixture of room air and test gas (94% $O_2$, 1% $SF_6$ and 5% $N_2O$) up to a total bag volume of 3L. Switching between air and bag was controlled by fast-operating pneumatic valves triggered at the end of expiration, under the direction of the operator. A carbon dioxide ($CO_2$) scrubber was placed in sequence between the bag and patient, so that expired air was depleted of $CO_2$ prior to re-inspiration. Initial bag volume was adjusted to be approximately equal to estimated FRC based on subject height, with test gas bolus of 30–40% total bag volume and balance room air. The bag volume and test gas bolus fraction could be increased if longer wash-in was required. At the start of wash-in, participants took 5–6 slow deep inhalations

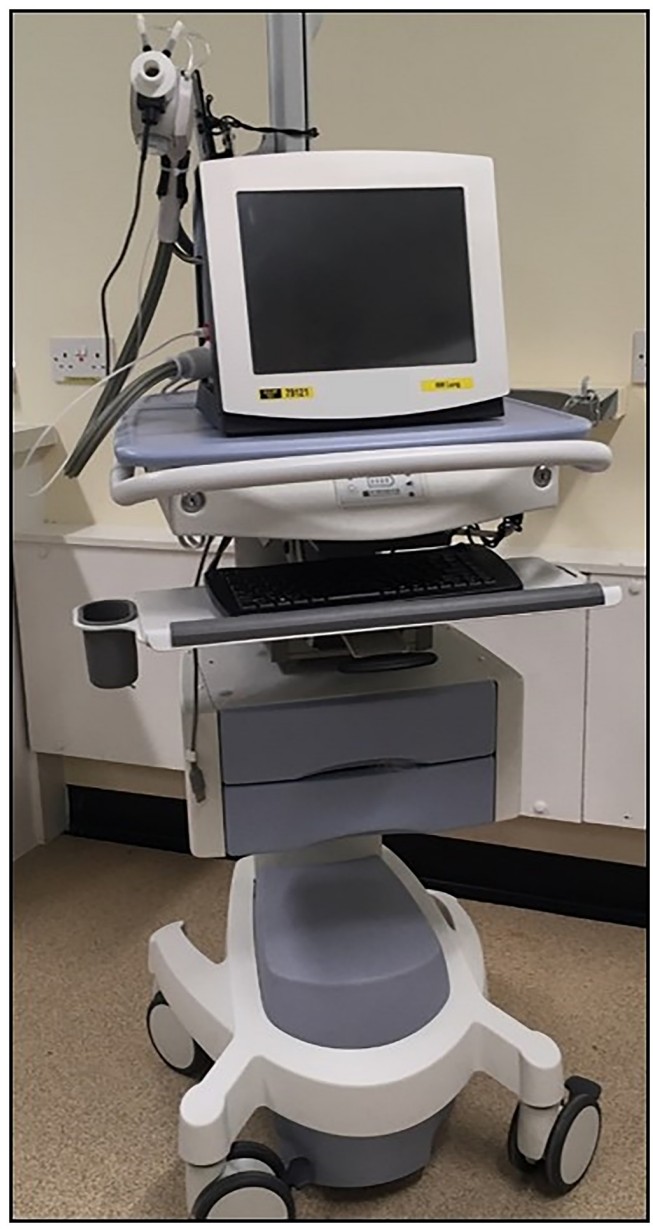

**Fig 1. Innocor closed circuit MBW system, mounted on medical cart with on-board power supply for portable MBW measurements.**

before returning to tidal breathing. Final washin concentration of expired $SF_6$ was between approximately 0.1 and 0.4%, depending on the starting concentration in the bag and the ratio of bag volume to FRC. It is assumed that LCI is independent of these small changes in tracer gas concentration. Once equilibrium was reached (difference between inspired and expired $SF_6$ concentrations <1%, adjusted for $CO_2$ removal), wash-in was continued for a further 30–60 seconds to ensure washin was complete, as previously described [22]. During wash-in, inspired $CO_2$ and $O_2$ concentrations were continuously monitored. Inspired $CO_2$ was typically <2% at end of wash-in, and inspired $O_2$>20%. At the end of wash-in, the participants were

switched to room air and instructed to maintain tidal breathing. Washout was continued until the expired end-tidal $SF_6$ concentration reached <2.5% of the starting concentration. Expired gas was dispersed by use of a fan directed at the patient and exhalation port. There was no requirement for a delay between end of washout and start of next wash-in, and subjects started the next test as soon as they were able (this is a feature of all exogenous MBW systems). Distraction was provided in the form of a screen showing age-appropriate movies or TV shows. In the case of adults, visual feedback of inspiratory volumes was available to aid reproducibility of breathing patterns, and typically set at 10-15ml/kg. Both children and adults used identical patient interfaces and mouthpieces, with the only difference being that a smaller filter was used in children (subjects <18yrs).

Subjects completed three washouts. If one or more tests were obviously compromised (e.g. evidence of leak), then additional tests were added. Detailed analysis and quality control were performed offline (see below). Following MBW testing, all adult volunteers and those children assessed at RMCH and WH also completed spirometry. This was not carried out in the school volunteers or at RSUH. Spirometry was performed using an Easyone handheld spirometer (NDD Medizintechnik AG, Zurich, Switzerland), according to ERS/ATS guidelines [23]. Normal ranges for spirometry were those from the Global Lung Initiative[24].

## Washout analysis

The Innocor device provides measurements of LCI and FRC, which was used in real-time to check test repeatability, but for this study a separate offline washout analysis package was used, prepared in-house (software version 6, release date 11/11/16) in Igor Pro v6 (Wavemetrics Inc., Lake Oswego, OR, USA). This is based on the same washout analysis package already deployed in several other clinical studies and clinical trials [4, 14, 25]. In this latest version, the alignment of flow and gas concentrations (performed as a daily calibration step) were checked against those actually measured during washout, and were adjusted to match in the event of differences between alignments of >10ms. Adjustment was also made for re-inspired $SF_6$, and cumulative expired volume was adjusted to account for total equipment deadspace, as per consensus guidelines [1]. FRC is quoted as FRC at the airway opening and was adjusted for pre-gas sampling deadspace (total 62ml adults, 58ml children, differing only in the choice of filter), for re-inspired SF6, and for BTPS.

The final LCI and FRC measurements quoted are the average of at least two reproducible repeats, as measured using the offline software package described above. Repeats were excluded if there was evidence of leak, or in case of large differences seen in LCI or FRC measurements (>25% from median) [1]. Washout test time was taken from the length of the washout file. This is the total time to complete all wash-in and washout tests, including any interval between tests, and analyser warm-up time (60 seconds). It does not include time taken to explain the test to the participants, or time taken to clean the apparatus between volunteers.

## Statistical analysis

Data were analysed using Prism (GraphPad Software Inc., San Diego, CA, USA) and R v3.0.2 (R Foundation for Statistical Computing, Vienna, Austria). Normal distribution was assessed using the Shapiro-Wilk normality test. Data are presented as mean (SD) or median (interquartile range [IQR]) unless otherwise stated. An upper limit of normal (ULN) corresponding to the 95th centile (i.e. mean +1.64 x standard deviation) was used as recommended by the Global Lung Function Initiative [24]. Repeatability of testing was defined by the within visit coefficient of variation of washout repeats (CoV), defined as the standard deviation divided by the mean. The final dataset contains LCI values derived from only two successful repeats, a

scenario in which the CoV may not be appropriate[1]. Values for CoV are therefore given for both the entire dataset as well as separately for those with all three measurements. Average LCI in different populations were compared using t-tests or ANOVA (for more than two groups). For comparison of different centres, the paediatric populations measured at the two Manchester sites by the same team (WH and RMCH) were merged (here described as LCI RMCH). For non-parametric data, Mann-Whitney U test was used to compare groups. Bland-Altman was used to compare measured to predicted FRC, using separate reference equation for adult[26] and paediatric populations[27]. Univariate regressions were used to examine the relationship between LCI and key demographics (age, gender, height, weight and BMI). Multivariable regressions were considered in the event of significant univariate findings. Due to the discrepancy between the numbers of adults and children and also the lack of those aged over 40, sensitivity analyses were performed for individual subsets (aged <18, aged≥18 and aged<40). A p-value <0.05 was considered to be statistically significant.

## Results

One hundred and seventy two subjects were recruited (82 males). The data from four subjects were not used due to equipment technical factors (n = 3) or clinical reasons (n = 1) and have not been analysed further. Of the remaining 168 subjects, 160 successfully completed LCI measurements (95%) and eight were excluded due to failure to produce measurements or poor repeatability. Demographic data are presented in Table 1. Overall there were only 5 ex-smokers, average 0.4 pack years (maximum 4 pack years). Due to its small size, this group have not been analysed separately.

### Success of LCI measurements

The eight participants who were excluded because they were unable to generate usable data were all children, median (IQR) age 10.0 (7.5–12.5) yrs.

144 subjects (90%) achieved a completed assessment with three washout repeats, 15 subjects required 1–3 additional repeat measurements at the same sitting. In two cases, for logistic or technical reasons, a third repeat was omitted.

**Table 1. Summary demographics of the study population.** Data are shown on the left for all subjects, including those with unsuccessful measurements, and on the right for the population used to derive the normal range (those with successful measurements, aged 5-39yrs).

| Study population | All included subjects | Subjects <40yrs with valid LCI |
|---|---|---|
| **Number of subjects** | 168 | 153 |
| **Male: Female** | 79:89 | 72:81 |
| **Median Age (range) yrs** | 13 (5–59) | 13 (5-39yrs) |
| *Median age (range, n) for ADULTS* | *29 (18–59), n = 52* | |
| *Median age (range, n) for CHILDREN* | *11 (5–18), n = 116* | |
| **Median BMI (IQR) (kg/m$_2$)** | 20.0 (17.2–23.1) | 19.8 (17.1–22.8) |
| **Median FEV$_1$ z score (IQR) L/s** | -0.24 (-0.88 to 0.31) (n = 81) | -0.26 (-0.88 to 0.25) (n = 75) |
| **Median FVC z score (IQR) L** | -0.25 (-0.77 to 0.28) | -0.27 (-0.72 to 0.25) |
| **Median FEF$_{25-75}$ z score (IQR) L/s** | -0.32 (-0.81 to 0.52) | -0.30 (-0.81 to 0.42) |
| **Mean FEV1/FVC (SD)** | 0.84 (0.06) | 0.85 (0.06) |
| **Mean LCI (SD)** | 6.13 (0.46) *(n = 160)* | 6.10 (0.42) *(n = 153)* |

FEV$_1$: forced expiratory volume in 1 second; LCI: lung clearance index; SD: standard deviation, IQR: interquartile range, BMI: body mass index, FVC: forced vital capacity, FEF$_{25-75}$: forced expiratory flow over 25–75% of expired volume.

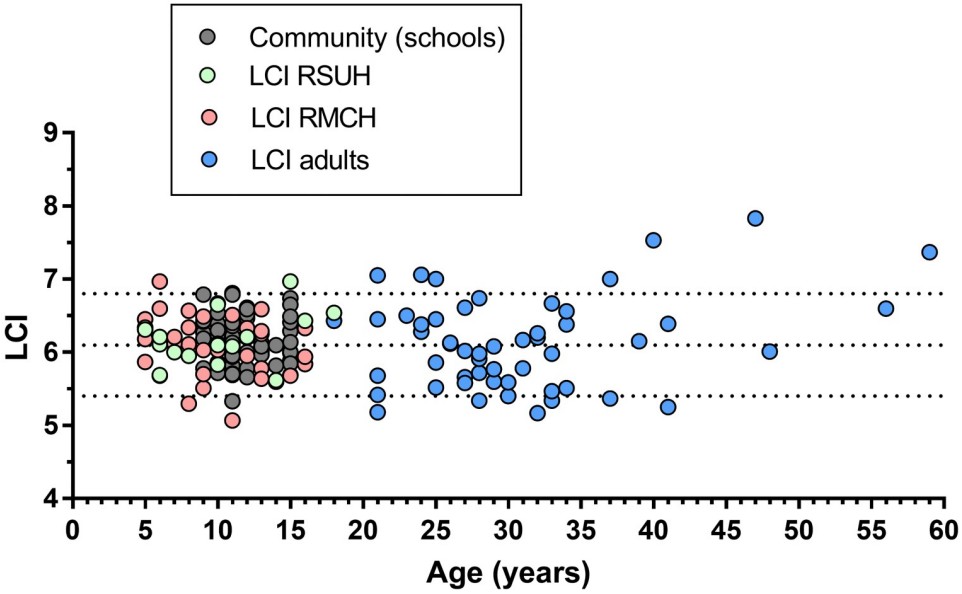

**Fig 2. Lung clearance index (LCI) measurements against age for healthy volunteers from the four centres.** Central dotted line represents mean, with upper and lower limits of normal shown by upper and lower dotted lines respectively. RSUH: Royal Stoke University Hospital; RMCH: Royal Manchester Children's Hospital.

58 individual washout repeats (11.6% of the total) were subsequently excluded due to insufficient quality control or repeatability. In the final analysis, 37 (24%) of subjects LCI values were derived from two repeats and 117 (76%) from three repeats.

Median (IQR) CoV for all included LCI measurements was 3.4 (1.8–5.2)%, or 3.3 (1.9 to 4.9)% for those with three measurements. Median CoV for all FRC measurements was 4.3 (2.4–6.3)%, or 4.3 (2.5–6.4)% for those with three measurements.

## Lung clearance index in healthy subjects

It was apparent from inspecting the data that LCI measurements in those aged over 40 years appeared to be more dispersed than in younger subjects. Although it seems likely that LCI may be higher and more varied in this older group, given the small number of subjects (n = 7) it is not possible make a confident statement regarding the LCI trend in those aged over 40 years. We therefore repeated the analysis, including only those aged under 40 years at the time of measurement. Mean (SD) for this cohort (n = 153) was 6.10 (0.42), making the upper limit of normal (ULN) 6.8 (see Table 1).

## Comparison of LCI between adults and children

Linear regression models showed no relationship between age and LCI in either the full dataset or the subsets of children or adults (β = 0.004, -0.005 and 0.019 respectively) (Fig 2). Mean (SD) LCI in children (<18yrs) was 6.13 (0.36), compared to an adult aged under 40 mean of 6.03 (0.53). Due to the wider SD in these adults, the ULN for children could be calculated as 6.72 compared to 6.90 in adults. However there are fewer subjects in the adult cohort (n = 46 vs 107), which may partially explain this difference. Given the minimal differences between the two calculations, and lack of age-related influence on mean LCI, we therefore propose 6.8 as the ULN for LCI (measured using this system) for all subjects aged between 5–39 years inclusive.

## Effect of other demographic variables on LCI

LCI was not significantly different between male and female subjects (mean difference = 0.074, 95% CI 0.071 to -0.219, p = 0.313) or between the paediatric measurements at the three different centres (p = 0.883). No relationship was found between LCI and height (β = -0.002, p = 0.427) across the entire population, Fig 3. This was also true if the adult and paediatric LCI populations were considered separately (β<0.001, p = 0.961; β = -0.003, p = 0.182, respectively). Likewise there was no association between LCI and BMI in either the full dataset (β 0.005, p = 0.467) or for the adult or paediatric populations separately (S1 Table in S1 Data). Spirometry data were incomplete (n = 81), but there were no significant correlations identified between any spirometric index (FEV$_1$ z score, FVC z score, FEF$_{25-75}$ z score, FEV$_1$/FVC) and LCI (all r$^2$<0.015, p>0.28).

## Functional residual capacity

The relationship between height and FRC is shown in Fig 4. As expected for a measure of lung volume, this increased exponentially with height from a minimum of 0.65L to a maximum of 4.76L (a 7 fold increase). FRC from washout was compared to predicted FRC[26, 27]. There was no significant difference between measured FRC from MBW and that derived from the prediction equations (median difference of 0.04L, p = 0.2). On Bland-Altman comparison, there was no evidence of consistent or size-related bias, with a mean (SD) difference of 0.06 (0.56) L, or 0.9% of predicted FRC, in favour of the prediction equations (see S1 Fig in S1 Data).

## Test time

Median (IQR) overall test time was 18.7 (16.0–22.5) minutes. Test time was shorter in children, with a median time of 17.8 minutes, compared to 23.5 minutes in adults (p<0.0001). There was a weak but statistically significant correlation of test time with age (r$^2$ = 0.29, p<0.0001) (Fig 5).

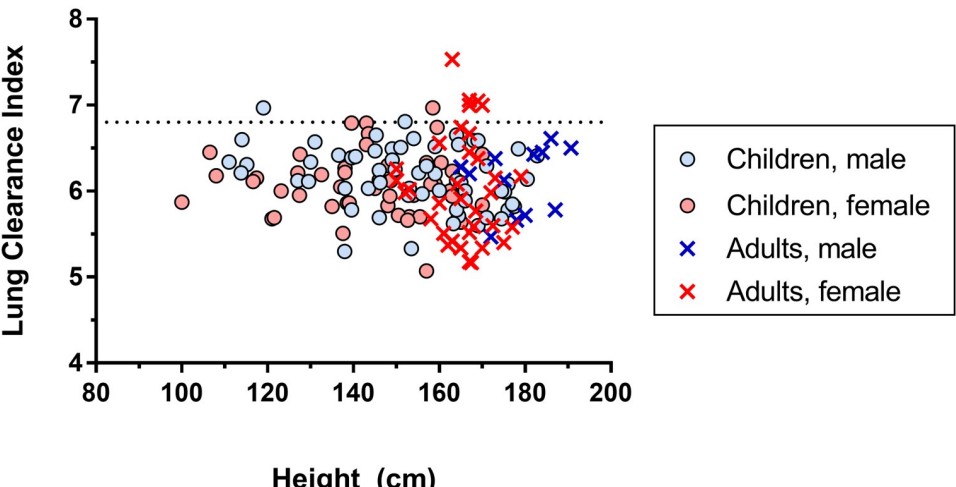

**Fig 3. Lung clearance index (LCI) measurements against subject height for healthy volunteers.** The upper limit of normal LCI is shown as a dotted line. Paediatric subjects are indicated by round circles, and adult subjects by crosses. Male and female subjects are indicated by blue and red circles symbols respectively.

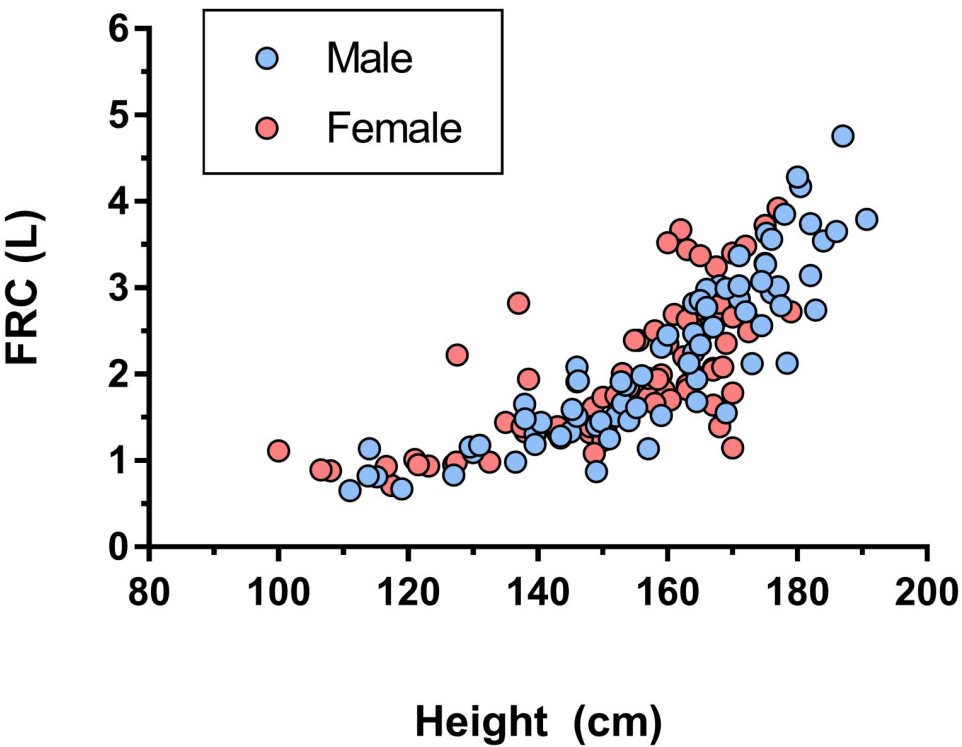

**Fig 4. Functional residual capacity (FRC) measurements against subject height for healthy volunteers.** Male and female subjects are indicated by blue and red circles respectively.

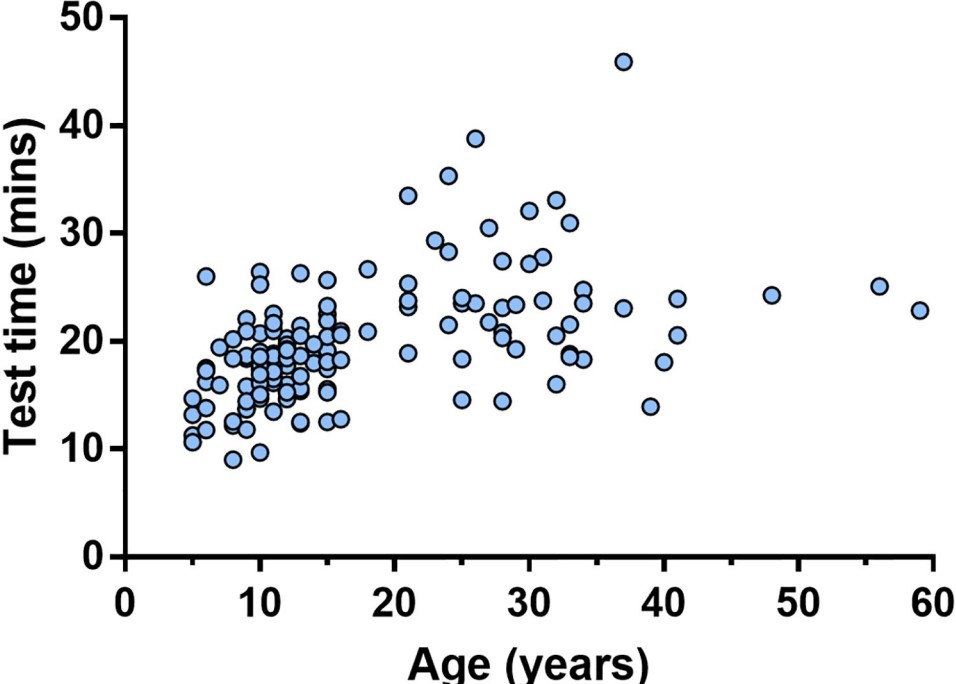

**Fig 5. Plot of total test time in minutes against age of subject (years).** Test time includes the time taken to complete all washout repeats, including those subsequently excluded from lung clearance index analysis, as well as the interval between washout repeats.

## Discussion

This is the first time such a large cohort of healthy volunteer data has been described using $SF_6$ with a closed circuit wash-in to achieve a more rapid and efficient test. These data will provide more accurate understanding of disease severity in clinical studies conducted using this method. The results highlight one of the strengths of LCI as an outcome measure using this method, as there is a stable range of normality across most of this age range. This is especially important for longitudinal paediatric studies, where large changes in lung volumes related to growth can make it difficult to identify disease-related change. With LCI, however, change over time can be confidently ascribed to disease processes or treatment effect. Finally, we have described for the first time the use of a genuinely portable system, operated by a single researcher in a community setting.

The range of normal described here is very similar to that for previous cohorts involving $SF_6$ as the tracer gas. However the mean and upper limit of normal are both lower in this study compared to those described with the same tracer gas in comparable populations. These range from an upper limit of normal LCI of 7.5 [12] to 7.2 [28] using a mass spectrometer (both in subjects up to 18 years old), and 7.4 using an Innocor system with the open circuit wash-in in subjects up to 58 years [11]. The current washout system however has been updated from that used in the earlier study using an Innocor device [11]. In particular, the patient interface has now changed so that, whilst total deadspace remains similar, the post-capillary deadspace has increased to 4.5ml. The analysis therefore now adjusts for re-inspired $SF_6$ volume, as well as adjusting the expired volumes for equipment deadspace (as recommended in guidelines that were not available when this earlier study was conducted) [1]. In addition, the method of signal alignment has been refined and standardised, so that the daily calibration is now checked against the alignment seen in washout and adjusted if this has not been correctly calculated. This method is significantly different from that previously described where the alignment was altered to artificially enhance the signal response time [11]. Thus, although the central methodology is unchanged, it is not unexpected that the final adjusted outputs, which align the analysis with the latest consensus guidelines, are slightly different from those described over 10 years earlier.

Unlike previous studies, we have also not seen a convincing change in LCI with age. There is an important caveat here however, namely that the number of subjects 40 years and older was very limited (n = 7). Previous studies have shown increases in LCI at either end of the age spectrum. Lum *et al.* described increased $SF_6$-LCI in very young controls, possibly due to the effects of deadspace and posture, but these subjects were younger than the lower limit of the current study [12]. Using an SF6 washout system, Fuchs *et al.* also did not show any relationship between LCI and age up to 20 yrs[29]. On the other hand, lung elasticity declines with age, and this is associated with increases in $FEV_1/FVC$, increased trapped gas (RV/TLC), and increases in a number of markers of ventilation heterogeneity including LCI[19]. Kjellberg *et al.* reported on nitrogen-LCI from 400 healthy adult controls up to the age of 71 years, measured using an Exhalyser D device [18]. Mean and ULN LCI were higher than that reported here and increased linearly with age from 17 years. More recently Anagnostopolou et al described nitrogen LCI from 180 children aged 6-18yrs [30]. Mean (SD) LCI was 7.04 (0.45). Although mean LCI increased with age (0.5 units over 12yrs), this was less than described by Kjellberg in older subjects [18] and, similar to our approach, the authors concluded that a fixed ULN was appropriate. Htun *et al.* and Verbanck et al have separately reported on nitrogen-LCI in similar populations using a different non-commercial nitrogen washout system [19, 31, 32]. The normal range for LCI in both cases was different, but in contrast to the Exhalyser D data, seemed to show relatively stable ULN for LCI below 40 years, with LCI only

increasing above this age. Verbanck *et al.* in particular showed rapid increase in LCI above about the age of 50 years, associated with an increasing spread of values [32]. Our own data show that, at least for the younger patients, $SF_6$-LCI is similar to nitrogen-LCI in having a stable ULN. We have few subjects above 40 years, but in this older age group the LCI data appear also to be more dispersed, with three measurements being above the group ULN. It seems likely that increasing ventilation heterogeneity is therefore a feature of ageing lungs, and a static ULN is unlikely to be applicable for older adults. It would therefore be sensible to view MBW results for the over 40 year olds with caution, and recommend that more work should be done in this age group.

In this study we have observed a wider spread of normal range data in adults, despite the mean LCI being almost identical between the two groups with no age-related dependence. The net effect of this small increase in data spread is to generate an ULN for LCI of 6.7 for children and 6.9 for adults. However, given the small differences between these values, and the smaller dataset in adults contributing to this spread, it is reasonable to propose a fixed ULN of 6.8 for the healthy subjects from 5-39yrs.

As expected, there was an exponential increase in lung volume (FRC) with height. Previously reported nitrogen-MBW systems have substantially over-estimated FRC *in vivo*[14, 17], despite convincing data on *in vitro* accuracy[33]. This error appears to be more pronounced in disease but has also been seen in healthy subjects[13]. The reasons for this are unclear, and may relate to the contribution of body nitrogen to the washout or to issues with specific analyser technologies[34, 35]. In this study however, we saw no consistent deviation in FRC values from those predicted, with a mean difference of only 0.06L from predicted FRC.

The other significant development described here is the use of LCI in a community setting. The use of a compact and robust analyser, combined with a gas source stored in small onboard cylinders, means that we were able to package the entire system into a transport case and take it directly into schools. This is important for community based assessments of MBW which could include measurements carried out in workplaces or family doctor surgeries. The small cylinders used in this study typically provide enough tracer gas for 5–10 sets of washout measurements (depending on subject size). The wash-in protocol deployed here allows for highly efficient wash-in within 2–3 minutes. Overall, the median time to complete all three tests was under 19 minutes, however this would likely be higher in populations with lung disease as more abnormal gas mixing would require a longer wash-in and washout [14]. Overall success rates were high, and that in children (93%) is higher than the success rate of 83% recently described in a multi-centre of four experienced LCI centres [30].

The data described here refer to the Innocor system and the use of $SF_6$ as a tracer gas. They cannot be used to identify normal ranges in subjects performing a nitrogen washout. Commonly deployed nitrogen washout systems seem to generate much higher values of LCI and FRC than $SF_6$ washouts [13, 14] due to differences in the behaviour of the tracer gases [13], the washout of body nitrogen during testing [35, 36], and the equipment itself[17]. The data analysis package was developed in-house, and is not the same as that shipped with the Innocor device. However it has been developed from that deployed in multiple previous studies[4, 37], including the CF gene therapy trial[25], and updated to incorporate consensus recommendations [1]. It has not been possible to compare LCI using our data analysis package with that of the Innocor system due to differences in flow-gas delay correction. All washouts in this dataset were reviewed and analysed by an experienced operator (AH).

In conclusion, we have described the first large scale use of the closed circuit washout method, in a sizeable cohort of healthy controls. We have established the baseline for normative LCI, and shown that this is stable across the age range 5–39 years. We have also shown

that it is possible to take MBW measurements out of a clinical setting, and that these can be conducted efficiently and reproducibly even in a community setting such as a school.

## Supporting information

**S1 Data.**
(DOCX)

## Acknowledgments

The authors would like to thank the pupils from Woodhouse Academy, Stoke on Trent, and the JCB Academy, Uttoxeter, who participated in this study as well as the staff and parents who facilitated this. This work was funded by the NIHR and supported by the NIHR Manchester Clinical Research Facility and NIHR Manchester Wellcome Trust Paediatric Clinical Research Facility. AH is also supported by the NIHR Manchester Biomedical Research Centre. This report therefore presents independent research funded by the NIHR. The views expressed are those of the authors and not necessarily those of the UK National Health Service, the NIHR or the UK Department of Health.

## Author Contributions

**Conceptualization:** Alex R. Horsley, Andrew Jones, Jaclyn Smith.

**Data curation:** Alex R. Horsley, Amnah Alrumuh.

**Formal analysis:** Alex R. Horsley, Amnah Alrumuh, Catherine Fullwood.

**Funding acquisition:** Alex R. Horsley, Jaclyn Smith, Francis J. Gilchrist.

**Investigation:** Alex R. Horsley, Amnah Alrumuh, Brooke Bianco, Katie Bayfield, Joanne Tomlinson.

**Methodology:** Alex R. Horsley, Steve Cunningham.

**Project administration:** Alex R. Horsley, Andrew Jones, Anirban Maitra, Anand Pandyan, Francis J. Gilchrist.

**Resources:** Alex R. Horsley, Francis J. Gilchrist.

**Software:** Alex R. Horsley.

**Supervision:** Alex R. Horsley, Andrew Jones, Anirban Maitra, Steve Cunningham, Jaclyn Smith, Anand Pandyan, Francis J. Gilchrist.

**Validation:** Alex R. Horsley.

**Visualization:** Alex R. Horsley.

**Writing – original draft:** Alex R. Horsley.

**Writing – review & editing:** Alex R. Horsley, Amnah Alrumuh, Brooke Bianco, Katie Bayfield, Joanne Tomlinson, Andrew Jones, Anirban Maitra, Steve Cunningham, Jaclyn Smith, Catherine Fullwood, Anand Pandyan, Francis J. Gilchrist.

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
