## [Decision Letter · Decision Letter 0]

31 Oct 2019

PONE-D-19-27084

Lung clearance index in healthy volunteers, measured using a novel portable system with a closed circuit wash-in

PLOS ONE

Dear Dr. Horsley,

Thank you for submitting your manuscript to PLOS ONE. After careful consideration, we feel that it has merit but does not fully meet PLOS ONE’s publication criteria as it currently stands. Therefore, we invite you to submit a revised version of the manuscript that addresses the points raised during the review process.

The most important points addressed by both reviewers include further clarification about the study population and age range included especially with regard to the results, its evaluation and interpretation. Furthermore, both reviewers ask for some clarifications concerning the device and the software used, which is of importance to make it clearly understandable to the reader and with respect to the potential influence on the results. Please answer the reviewer's comments accordingly.

We would appreciate receiving your revised manuscript by Dec 15 2019 11:59PM. To enhance the reproducibility of your results, we recommend that if applicable you deposit your laboratory protocols in protocols.io, where a protocol can be assigned its own identifier (DOI) such that it can be cited independently in the future. For instructions see: http://journals.plos.org/plosone/s/submission-guidelines#loc-laboratory-protocols

We look forward to receiving your revised manuscript.

Kind regards,

Sophie Yammine

Academic Editor

PLOS ONE

Journal Requirements:

1. Please clarify whether ethical approval was granted by University of Manchester with Lancaster Research Ethics Committee and please provide affiliation of the latter. Please also include in the Methods section participants recruitment dates.

2. Thank you for including your competing interests statement; "AH reports a collaboration agreement with Innovision ApS.

All other authors report no conflicts of interest."

Reviewers' comments:

Reviewer's Responses to Questions

**Comments to the Author**

1. Is the manuscript technically sound, and do the data support the conclusions?

Reviewer #1: Partly

Reviewer #2: Yes

2. Has the statistical analysis been performed appropriately and rigorously? 

Reviewer #1: Yes

Reviewer #2: I Don't Know

3. Have the authors made all data underlying the findings in their manuscript fully available?

Reviewer #1: No

Reviewer #2: Yes

4. Is the manuscript presented in an intelligible fashion and written in standard English?

Reviewer #1: Yes

Reviewer #2: Yes

5. Review Comments to the Author

Reviewer #1: PONE-D-19-27084

LUNG CLEARANCE INDEX IN HEALTHY VOLUNTEERS, MEASURED USING A NOVEL PORTABLE SYSTEM WITH A CLOSED CIRCUIT WASH-IN

In this manuscript Horsley et al describe MBW measurements from a substantial cohort of healthy children and adults using closed-circuit methodology to reduce overall test-time with a portable SF6 based MBW system. The authors should be congratulated for their novel work to address lengthy MBW test-time and space limitation, both of which are significant practical barriers to integration of MBW into routine clinical practice.

To my knowledge this is the first time successful use of a portable MBW device in the community has been reported. Additionally the authors report shortened test time using a closed-circuit and deep breathing to achieve rapid wash in, which appears to be feasible and repeatable within a single test occasion across the healthy cohort.

My main concerns with this study relate to the evaluation and interpretation of normative data. The authors conclude that they have established “the baseline for normative LCI” and demonstrated a stable ULN in subjects between 5-40 years, which in my opinion is not fully supported by the data as presented.

My primary concerns relate to the following:

1. ULN or Reference Data Set?

As written, the main message related to the description of this healthy data set is somewhat unclear. The title and the aims of the study suggest this is an evaluation of their healthy control group and derivation of an ULN ahead of a local longitudinal clinical study. However, a number of places throughout the manuscript, particularly in the discussion, imply that this data is generally applicable to ”provide more accurate understanding of disease severity in clinical studies conducted using this method ” (P. 10) and “establish the baseline for normative LCI”. Please clarify whether this is intended to be the first step towards developing a reference data set for this gas/device or simply to report ULN in this healthy cohort.

2. MBW results are device/software specific.

In addition to being gas specific, differences in hardware, test methodology, data acquisition, software settings, calculation of outcomes, and analysis algorithms etc. all have the potential to impact LCI. I strongly disagree that the data analysis software does not impact MBW outcomes generated as is currently stated in the discussion (P.13) Previous groups have shown that MBW outcomes are significantly impacted by software (Summermatter S, et al PLoS One 2015: 10: e0132250, Anagnostopoulou P, et al Pediatr Pulmonol 2015: 50: 970-977) used to calculate results. While the consensus guidelines provide some measure of guidance there are many device specific details that are equally impactful. Strongly suggest to clearly acknowledge these limitations and clearly state that these results are are device and data analysis software version specific.

Suggest the authors consider the following:

• P6 L154: Please clearly state whether the results reported in this manuscript were derived from the Innocor device or the separate offline analysis software.

• Is the offline analysis software commercial? Please include device software version and offline analysis software version in the methods.

• Has the device, in its current form, been validated as recommended in the MBW consensus guidelines (Robinson et al, ERJ 2013)?

• The authors have indicated that FRC is calculated to the airway opening but have not specified calculation of LCI, ie was CEV corrected for full dead space or just re-inspired tracer gas?

• How do the offline results compare to the device calculations of MBW outcomes?

3. Is a single fixed ULN appropriate for the entire age range?

The authors aim to demonstrate that there is no linear relationship between LCI and age or height in the overall healthy cohort however, the data as presented do not clearly lead the reader to reach the same conclusions. There are some open questions that should first be addressed.

a) The authors cite previously published work as reporting a stable ULN for LCI (SF6) in healthy children (Lum et al, Eur Respir J 2013: 41: 1371-1377), a statement which could be mis-interpreted by an unfamiliar reader. Lum and colleagues report an association of LCI with height that, while minimal from approximately age 6 (~120cm), is present throughout childhood. They do suggest that a fixed ULN may be appropriate for cross-sectional analysis but caution against using a fixed upper limit for serial measurements at any age during childhood as LCI continues to change with development.

Additionally, technical and physiological factors, such as equipment specific mechanical dead space (Benseler A, Respirology 2015: 20: 459-466) and tidal volume (Yammine S, J Cyst Fibros 2014: 13: 190-197, Ratjen F, Plos One, ISSN 1932-6203, 07/2019, Volume 14, Issue 7) can dynamically influence LCI as the child grows. These effects will depend somewhat on the device and the specific population being tested and should be ruled out.

Suggest the authors consider the following:

• Separate evaluation of the pediatric data to determine whether there is an age/height dependency for LCI in the pediatric group. If association is found to be present, discuss the rationale for using a fixed ULN rather than prediction equations to express LCI as a z-score to account for growth in the setting of a longitudinal study. If the magnitude of the effect over the age range is small a fixed ULN may still be appropriate.

• Is the association between LCI and age/height/BMI different between adults and children?

• While mean LCI was not different between adults and children, suggest to confirm that the ULN is also unchanged.

• Have the authors considered the impact of equipment dead space on LCI? Particularly in children where recommended ratio of Vd to weight is no more than 2 ml/kg, total reported Vd here is 58ml making min weight 29 Kg.

b) In the adult population, the authors report conflicting evidence of increase in LCI with age in previously published papers with adults 17 and older (Kjellberg et al, Htun et al. and Verbanck et al.) derived from MBW using N2 as tracer gas. The authors state they have not observed a convincing increase in LCI with age in the overall adult cohort (n= 52) but acknowledge that a fixed ULN is likely inappropriate for adults over 40 (n=6) which is supported by Fig 2.

• What was the rationale for choosing age 40 as the upper cut-off in this data set?

• If the authors feel that more data is required to evaluate LCI in older adults it would be prudent to confirm that the ULN is not different when subjects >40 years are excluded.

• Given there are only 7 subjects over 40 years, and the authors suggest these data should be interpreted with caution, the first statement of the discussion may be somewhat over-stated.

Specific comments by section:

Introduction

• P3. L48: consider additional pediatric references

• P3. L55: define SF6

• P3. L60-63: test time can differ with population, i.e. longer with very young children, these are generalized statements suggest to remove specific times and just point out that Spirometry is generally much faster than MBW.

• P4. L69- 72 Please briefly elaborate on how closed circuit methodology achieves faster wash-in

Methods

Study Design and Recruitment

• P5L95: States “ transported in a protective case and taken to two primary schools”, previously stated primary and secondary school – please clarify.

Study Assessment

Description of test procedure is unclear, suggest to edit for clarity (specific suggestions below)

• P6 L119: “Wash-in was performed from a sealed bag filled with a mixture of room air and test gas (94% O2, 1% SF6 and 5% N2O) up to a volume of 3L.” please clarify that this is total volume.

• P6 L123: “Initial bag volume was adjusted to be approximately equal to estimated FRC, with test gas bolus of 30-40%.” Please clarify that this refers to fraction of total bag volume.

• P6 L123: “These parameters could be increased if longer wash-in was required” Please clarify what parameters are referred to and why might they be adjusted?

• P6 L124: Why 5-6 deep breaths?

• P6 L125: “Once equilibrium was reached” What was absolute starting concentration?

• P6 L127: “…wash-in was continued for a further 30-60 seconds.” Why?

• Did all subjects use the same testing interface? Subsequently refer to different filters used-please clarify.

• P6 L137: Please describe “… detailed analysis and quality control”, what specific guidelines for quality control were used?

• P6 L132: “There was no requirement for a delay between end of washout and start of next wash-in, and subjects started the next test as soon as they were able.” This is somewhat misleading - this is a consequence of using exogenous gas for MBW not specific to this device.

• P6 L134: “In the case of adults, visual feedback of inspiratory volumes was available to aid reproducibility of breathing patterns” What was target Vt?

Washout Analysis

• Please clarify that offline software was used to calculate MBW outcomes reported here.

• P6 L151-152: The authors state that FRC is corrected for dead space and reported at the airway opening, but that CEV is adjusted to account for re-inspired gas only, please clarify that LCI is therefore reported to the gas sampling point. Please confirm that FRC used in calculation of LCI was also calculated to gsp.

• P6 L154: Please elaborate on quality control criteria used and rationale for using reproducibility as acceptability requirement for LCI, including references as appropriate. To my knowledge repeatability of LCI is not necessarily a criteria for disregarding data.

Statistical Analysis

• How was correlation of LCI with anthropometric parameters assessed? Were any relationships between MBW outcomes and physiological factors considered? (Vt/FRC, Vd/Vt etc)

• P8 L165: Within test variability of LCI and FRC was calculated using CoV. Please provide equation. Is CoV appropriate metric when only 2 washout repeats?

• Did the authors assess whether there was any difference between ever/never smokers in adults?

RESULTS

The result as presented are somewhat difficult to follow, suggest to edit for clarity to avoid confusion.

• Suggest clearly separating test performance (# tests/repeats attempted) from test success (# tests acceptable after QC). Possibly a flow diagram might help

• P9 L185-189: Is the intention to report number of subjects who required additional (more than 3) attempts to achieve a successful test?

• Please indicate the proportion of tests with 2 vs 3 successful repeats after quality control

• Please comment on the use of repeatability of LCI/FRC as QC criteria as it relates to reporting within-test variability in the results.

• P9 L212: Please include rationale for not evaluating relationship between LCI and BMI and FEV1/FVC in children? Please indicate data not shown or direct the reader to the results.

Discussion

• P11 L233: The ULN reported here is similar but lower than any previously reported in the literature using the same gas type, does the rapid technique ensure complete wash-in?

• P12 L256: Please clarify that Kjellberg et al. reported LCI derived from nitrogen based MBW

• Please ensure that all data reported in the results are mentioned in the discussion, how do the authors interpret the FRC results? The relationship between LCI and BMI or FEV1/FVC? How should the FEV1/FVC results in adults be interpreted in context of the aging lung?

General Comments:

Suggest the authors revise the language to improve overall readability

Figures:

• please ensure consistent nomenclature

• adjust scale so all data points are clearly visible

Fig 3: suggest presenting different symbols for peds and adults

Tables:

Table 1**Please ensure parameters reported in results are described in demographics**

• Please include anthropometric data (ht, wt/BMI)

• Please clearly indicate age cut-off for adults and children and check that this is consistent throughout the manuscript see P.9 L192.

• Suggest to also include

o number of ever/never smokers

o FRC z-score

o FEV1/FRC

Reviewer #2: Horsley and colleagues present an interesting study about multiple breath washout measurements in a healthy population using a portable device based on an established MBW setup. The study is straight forward and presented in a concise way. However, there is some statistical analysis missing, which would strengthen the results.

Please find my comments below:

1. The introduction starts with a paragraph regarding LCI use, but the number of references is quite limited and not fully updated. It is important to include more references, especially recent ones, in order to catch the eye of the average reader.

2. Introduction, line 56: Lum et al (ERJ 2013) showed also age-dependency of the LCI values, especially in infants and pre-schoolers, measuring with a mass-spectrometer using SF6.

3. Line 60-63: Are these data based on a publication, or is it based on the authors’ personal experience? This should be clearly stated.

4. Line 63-66: Reference 16 refers to an inexperienced population of young CF patients. The authors state that CF children with experience in the test take 1-6 min to perform a single trial.

5. Line 81: ‘…across a clinically-relevant age range’: what does this mean? Are the ages 5-59 clinically relevant? And, if so, why subjects below 5, or older than 59 are not relevant?

6. Line 123: Which parameters where used to estimate the FRC?

7. Line 128: What was the maximum inspired CO2 concentration during the washin? Did it influence the breathing pattern? Did the subjects keep a tidal breathing pattern during the washin time?

8. Did the authors perform any kind of power analysis? How did they come up with these numbers in children and adults? Was it in purpose to use less adults than children?

9. Basic demographics (weight, height, BMI, etc) are missing from Table 1.

10. Table 1: Which reference values were used for FEV1? The range is very high, and some FEV1 values seem to be very low. How do the authors comment on that, especially in a healthy cohort?

11. How was the CoV calculated?

12. In order to investigate the influence of demographics or other factors on the LCI, several studies have used more sophisticated statistical methods, like regression models. What are the effects of age, height, sex, etc on the LCI, using a regression analysis? What other factors could potentially influence this outcome?

13. The overall time of the test duration, although interesting, does not give exact information about the expected duration of a single trial, especially in children, where intervals between trials can be larger. Is there any way to estimate the duration of a single trial?

14. Although LCI is the primary outcome of the study, FRC calculation is critical for the estimation of the LCI. Apart from figure 4, there is no analysis showing effects of demographics and/or other factors on FRC. How do the authors discuss their findings in FRC in comparison to previous studies? Is the FRC critical for the duration of the test?

15. It would worth to mention and compare the results with other studies reporting reference values for LCI, either with the same (e.g. Fuchs S and colleagues, Pediatric Pulmonology 2009) or with other tracer gases.

6. PLOS authors have the option to publish the peer review history of their article (what does this mean?). If published, this will include your full peer review and any attached files.

Reviewer #1: No

Reviewer #2: No

---

## [Author Response · Author response to Decision Letter 0]

16 Dec 2019

A detailed response to the editors and reviewers extensive comments has been uploaded separately as "Response to reviewers", as requested by the editor.

---

## [Decision Letter · Decision Letter 1]

15 Jan 2020

PONE-D-19-27084R1

Lung clearance index in healthy volunteers, measured using a novel portable system with a closed circuit wash-in

PLOS ONE

Dear Dr. Horsley,

Thank you for submitting your manuscript to PLOS ONE. After careful consideration, we feel that it has merit but does not fully meet PLOS ONE’s publication criteria as it currently stands. Therefore, we invite you to submit a revised version of the manuscript that addresses the points raised during the review process.

The manuscript has gained in clarity and markedly improved, as stated by both reviewers. There remain some minor comments, two points of clarification about methodological aspects for reviewer 2, and for reviewer 1 regarding data presentation and analysis.

We would appreciate receiving your revised manuscript by Feb 29 2020 11:59PM. To enhance the reproducibility of your results, we recommend that if applicable you deposit your laboratory protocols in protocols.io, where a protocol can be assigned its own identifier (DOI) such that it can be cited independently in the future. For instructions see: http://journals.plos.org/plosone/s/submission-guidelines#loc-laboratory-protocols

We look forward to receiving your revised manuscript.

Kind regards,

Sophie Yammine

Academic Editor

PLOS ONE

Reviewers' comments:

Reviewer's Responses to Questions

**Comments to the Author**

1. If the authors have adequately addressed your comments raised in a previous round of review and you feel that this manuscript is now acceptable for publication, you may indicate that here to bypass the “Comments to the Author” section, enter your conflict of interest statement in the “Confidential to Editor” section, and submit your "Accept" recommendation.

Reviewer #1: (No Response)

Reviewer #2: (No Response)

2. Is the manuscript technically sound, and do the data support the conclusions?

Reviewer #1: Yes

Reviewer #2: Yes

3. Has the statistical analysis been performed appropriately and rigorously? 

Reviewer #1: Yes

Reviewer #2: Yes

4. Have the authors made all data underlying the findings in their manuscript fully available?

Reviewer #1: No

Reviewer #2: Yes

5. Is the manuscript presented in an intelligible fashion and written in standard English?

Reviewer #1: Yes

Reviewer #2: Yes

6. Review Comments to the Author

Reviewer #1: PONE-D-19-27084

LUNG CLEARANCE INDEX IN HEALTHY VOLUNTEERS, MEASURED USING A NOVEL PORTABLE SYSTEM WITH A CLOSED CIRCUIT WASH-IN

The authors have responded to reviewer comments and correspondingly revised the text to markedly improve the quality and clarity of this manuscript.

More thorough evaluation of their data has provided additional results to allow the reader to more fully appreciate the authors’ interpretation of their normative data. The additional analyses and methodological details now included in this manuscript further allow the reader to better understand how measurements were collected, and how outcomes and results were derived, in order to independently judge how derived ULN may be generalizable to other data sets.

Minor corrections/questions:

1. Reviewer Comment: P6 L125: “Once equilibrium was reached” What was absolute starting concentration?

Author Response:There was not a fixed starting concentration. This would depend on the bolus fraction, bag volume and lung volume. Equilibrium was established between end washin inspired gas and end washin

expired gas.

Reviewer Comment (R1): Please indicate that there was no fixed starting concentration. How much did starting concentration vary within a subject? Between subjects? Do the authors feel that this feature of the test may have impacted the results?

2. Reviewer Comment: P6 L127: “…wash-in was continued for a further 30-60 seconds.” Why?

Author Response:This was to ensure confidence in washin completion. Previously reported results have shown that it is possible to reach equilibrium very quickly with this method, but that in diseased lungs washin may

not be entirely complete. The method therefore requires to reach equilibrium and then continue

beyond this point.

Reviewer Comment (R1): Suggest to include rationale in text.

Reviewer #2: The authors have adequately responded to the comments. However, there are still some parts of the manuscript that could be improved.

1. In the abstract, the authors report the whole number of subjects with successful LCI measurements (age range 5-59) and later the mean and ULN LCI for the analysis cohort (5-39 years). For the average reader, this is confusing. I would suggest either to omit the age range for the whole number of subjects, or to transform the whole sentence in a way that it becomes clear which age range those reference values are applicable to.

2. Line 122: It is important to mention in the manuscript that the estimated FRC is based on height.

3. Table 1: The first column at table 1 refers to the whole number of participants (168) but not all of them had a successful MBW test. Thus, the mean LCI value does not apply to this number.

4. Line 205: This statistical comparison is not entirely correct, as it compares a group of 8 subjects with a group of >100 subjects, and it does not add anything to the results.

5. Line 242: Spirometry data were incomplete. How did the authors treat those data? By imputation?

6. Line 249: FRC measured vs predicted: Could the authors show the statistics? Is this difference significant?

7. Line 323: The mean difference of 0.06 L is not very informative for a parameter that is age and height dependent, such as the FRC. Please use the percentage instead.

8. Line 342: The impact of different analysis softwares on MBW results has been previously shown, as indicated in several studies, and mentioned previously by Reviewer 1. This statement is not in accordance with the response of the authors to the relevant comment.

9. Age influence on LCI: There is a new multicentre study showing reference values for nitrogen LCI in children, where, although LCI increased with age, this increase per year was minimal, and practically negligible (Anagnostopoulou et al, Eur Respir J 2019). The authors could consider this report when discussing stable upper limits of normal for LCI in children.

10. It should be included in the limitations of the study that the comparison with the Innocor analysis results was not feasible.

11. A table with the results of the univariate analysis would be useful.

7. PLOS authors have the option to publish the peer review history of their article (what does this mean?). If published, this will include your full peer review and any attached files.

Reviewer #1: No

Reviewer #2: No

---

## [Author Response · Author response to Decision Letter 1]

31 Jan 2020

Dear Dr Yammine

Thank you again to the reviewers for their feedback on this manuscript.

We have uploaded a response to each of the reviewer comments, along with a revised manuscript. These were mostly minor corrections or deletions but, as requested, we have also provided some more data. We have placed this in an additional supplement for the interested reader.

We hope that these changes will prove satisfactory.

---

## [Editor Report · Decision Letter 2]

4 Feb 2020

Lung clearance index in healthy volunteers, measured using a novel portable system with a closed circuit wash-in

PONE-D-19-27084R2

Dear Dr. Horsley,

We are pleased to inform you that your manuscript has been judged scientifically suitable for publication and will be formally accepted for publication once it complies with all outstanding technical requirements.

With kind regards,

Sophie Yammine

Academic Editor

PLOS ONE

Additional Editor Comments (optional):

One minor Reviewer comment (nr 7 of Reviewer 2) was probably overlooked. I leave it up to you, to add the equivalent % of predicted FRC (p.15, line 325).

---

## [Editor Report · Acceptance letter]

13 Feb 2020

PONE-D-19-27084R2 

Lung clearance index in healthy volunteers, measured using a novel portable system with a closed circuit wash-in 

Dear Dr. Horsley:

I am pleased to inform you that your manuscript has been deemed suitable for publication in PLOS ONE. Congratulations! Your manuscript is now with our production department. 

With kind regards,

on behalf of

Dr. Sophie Yammine 

Academic Editor

PLOS ONE